# Insights on the Formation Mechanism of Ultra-Low Friction of Phenolic Resin Graphite at High Temperature

**Fan Zhang** [1,*] **, Peng Yin** [1] **, Qunfeng Zeng** [2,*] **and Jianmei Wang** [1]

1 Engineering Research Center of Heavy Machinery of Ministry of Education, Taiyuan University of Science and Technology, Taiyuan 030024, China; yinpeng19981021@163.com (P.Y.); wjmhdb@163.com (J.W.)
2 Key Laboratory of Education Ministry for Modern Design and Rotor-Bearing System, Xi'an Jiaotong University, Xi'an 710049, China
* Correspondence: zhangfan19860605@126.com (F.Z.); qzeng@xjtu.edu.cn (Q.Z.)

**Abstract:** In the present paper, the influences of high temperature on the tribological properties of phenolic resin graphite (PRG) sliding against tungsten carbide-nickel (WC-Ni) alloy in ambient air were investigated systematically. Results demonstrated that the antifriction behaviors of PRG was sensitive to high temperature and PRG exhibits ultra-low coefficient of friction (CoF) of about 0.01–0.015. The low CoF is attributed to the formation of graphite tribofilms, which shows different formation processes on the contact interface at different temperatures (room temperature, 200, 300 and 400 °C). These findings provide insight into the formation mechanism of graphite tribofilms, and provide an important basis for improving the tribological properties of graphite-based friction materials and manufacturing new graphite for seal applications.

**Keywords:** phenolic resin graphite; ultra-low friction; graphite tribofilms; WC-Ni alloy

## 1. Introduction

Performance of the mechanical seal in an aerospace engine is mainly determined by the tribological properties of friction pairs which are mostly composed of graphite and cemented carbides [1,2]. During the use of the mechanical seal, due to excessive frictional temperature rise, the surface of the metal moving ring is deformed at high temperature and blue due to overheating. Moreover, the graphite ring is excessively worn and there is abnormal wear such as furrows. Therefore, it is necessary to study the friction and wear performance of the pair at high temperature. Compared with traditional basic lubricants such as molybdenum disulphide (MoS$_2$) [3] and polytetrafluoroethylene (PTFE) [4], graphite has been recognized as the ideal seal material and is commonly used as a soft face in the friction pair of the seal due to its self-lubricating performance, high thermal conductivity, excellent chemical stability and corrosion resistance [5]. The hard face is often composed of tungsten carbide (WC)-based cemented carbides because of their mechanical strength, high abrasion and corrosion resistance [6,7].

Generally, the structure, shape and size of graphite influence the friction and wear behaviors [8], but the tribological properties of graphite depend on not only the crystal structure of graphite, but also the manufacturing process [9]. The traditional manufacturing process of graphite forms pores in its matrix and on its surface. This is to improve the mechanical strength of graphite and expand its application with the optimized tribological performance under any sealing conditions and invulnerable to the friction-induced heating and excessive wear [10,11]. The porous graphite can be impregnated with metal, polymer, or inorganic salt. Impregnated graphite has attracted considerable attention because of its better tribological properties than non-impregnated graphite. Zhu et al. [12] studied the friction and wear behavior of resin graphite composite using a pin-on-disc configuration under a dry sliding condition. They have found that the resin graphite composite exhibited much better tribological properties than the non-impregnated graphite, and the steady

transferred film was easily formed on the counterpart surface because of the interaction of furan resin and wear debris of graphite. Jia et al. [13] experimentally studied the tribological properties of impregnated graphite under a dry and corrosive environment, and the results showed that the degree of graphitization had an important influence on the friction coefficient. Hirai et al. [14] explored the friction and wear characteristics of antimony-impregnated-carbon-graphite and studied the transferred films from seals to mating metal surfaces under dry, water and steam lubrication conditions. Zhang et al. [15] researched the friction and wear behavior of resin-impregnated graphite against WC-Ni alloy under dry, oil and water environments, finding that the impregnated graphite exhibited much better friction properties under water or oil lubrication than non-impregnated graphite, and there was wear phenomenon at the WC-Ni surface, which was due to the hard WC particles embedded in the graphite surface [16]. These studies are mainly based on the following characteristics: graphite has a lamellar structure with weak interplanar van der Waals forces, which allows the formation of a carbon-based friction film during the friction process and effectively protects the friction interface [17–19].

The formation of the transferred film is strongly affected by the sliding speed of friction pairs and the temperature [20–23]. The tribological properties of impregnated graphite have been widely studied under various friction and environment conditions. However, the research on the mechanism of achieving ultra-low friction at high temperatures is insufficient. Graphite has poor oxygen resistance in the ambient atmosphere at high temperatures [24,25], which leads to its easy oxidation to carbon monoxide above 400 °C, and to carbon dioxide above 500 °C. The oxidized abrasive particles will change the friction and wear characteristics of graphite. For high-reliability mechanical seals, it needs to be better explored on the influence of temperature on the tribological properties of impregnated graphite and the formation mechanism of the transferred tribofilm [26]. Huai et al. [27] developed a graphite-based solid lubricant filled with $SiO_2$ and studied the tribological properties with a $Si_3N_4$ ball at 700, 800 and 900 °C. This solid lubricating coating displayed a stable adhesion to the substrate and a superior lubricant performance with a friction coefficient of about 0.05 at high temperature under air atmosphere. Kumar et al. [28] studied the tribological properties of turbostratic graphite against a 100Cr6 steel ball under high temperature in an ambient atmosphere. The research explored that a low friction coefficient in 2D structured graphite was mainly due to the passivation of the dangling bonds by high adsorbed oxygen, and the increase in friction coefficient was ascribed to the gradual loss of residual passivating chemicals due to tribochemical reactions. Lafon-Placette et al. [29] studied the friction and wear behavior of SiC/SiC and SiC/C pairs with different impregnated carbon under dry friction and a temperature of 120 °C, the wear mechanisms were found to be driven by the cracking process and oxidation. Zhao et al. [17] found that the friction coefficient and wear depth of impregnated graphite decreased with an increase in the friction temperature (20–160 °C). Furthermore, with the increase in temperature, the tribofilm adsorbed on the friction interfaces, became more uniform and stable, and its structure became more ordered during friction.

At present, the research on the tribological characteristics and mechanism of graphite under high temperature conditions are not systematic, especially on the stability of ultra-low friction coefficient under high temperature conditions. In this work, the tribological properties of phenolic resin graphite against tungsten carbide-nickel (WC-Ni) at room temperature (RT), 200, 300 and 400 °C in ambient air were investigated systematically. The ultra-low friction coefficient with a certain duration was obtained. Microscopy observations, Raman spectroscopy and elemental analysis were conducted on the graphite pins and the friction layer built upon the surface of WC-Ni alloy, to obtain a solid interpretation to the evolution mechanism of the tribological properties of graphite.

## 2. Materials and Methods

### 2.1. Materials

The impregnated graphite was impregnated with phenolic resin, the porosity of the PRG used in this work was about 5 vol.%, and the graphitization of the PRG was about 50–75%. Physical characteristics of the PRG are listed in Table 1. The PRG materials were provided by Xiangjiu Graphite Co., Ltd., Xingtai, China. Cemented carbide was fabricated by compacting and sintering WC and Ni powders, the compaction pressure was greater than 2200 MPa and the maximum sintering temperature was 1450 °C. The mass fraction of Ni binder is 10%. The main physical properties of the cemented carbide are also shown in Table 1. The WC-Ni cemented carbide materials were provided by Ningbo Vulcan Mechanical Seals Manufacturing Co., Ltd., Ningbo, China.

**Table 1.** Mechanical and physical properties of test materials.

| Material | Roughness, Ra(nm) | Young's Modulus (Gpa) | Hardness | Density (g/cm³) | Porosity (vol.%) | Graphitization (%) |
|---|---|---|---|---|---|---|
| WC-Ni | 13 | 550 | 1500 HV | 15 | \ | \ |
| PRG | 500 | 26 | 75 HS | 1.88 | 5.0 | 50–75 |

### 2.2. Friction Tests

The tribological properties of PRG against WC-Ni cemented carbide at room temperature, 200, 300 and 400 °C in ambient air were carried out on a pin-on-disc high temperature tribometer (Rtec instruments, San Jose, CA, USA). The schematic diagram of sliding friction test and the photo of friction pairs are illustrated in Figure 1. The PRG with a diameter of 6 mm and height of 12 mm was used as a static friction pair, and the WC-Ni cemented carbide with a diameter of 30 mm was used as a rotating friction pair. The friction surfaces of all friction pairs were polished by a brass lapping machine with diamond slurry, and the surface roughness of PRG and WC-Ni were about 500 and 13 nm, respectively. The friction pairs were cleaned ultrasonically in acetone and then blown by dry air before the tribotests and measurements. The test processes proceeded as follows: the pair samples were automatically heated to various target temperatures (RT, 200, 300 and 400 °C), and then kept at each target temperature for at least 45 min. Then the friction experiment started simultaneously with the onset of a normal load of 10 N and a speed of 100 mm/s. The surface topography and elements of the samples were investigated by scanning electron microscopy (SEM, su-8010, Hitachi, Tokyo, Japan). The element transfer and chemical changes of the surfaces were observed by Raman spectroscopy (LabRAM HR Evolution, HORIBA Scientific, Villeneuve d'Ascq, France), which is an effective method to detect the molecular structure of materials.

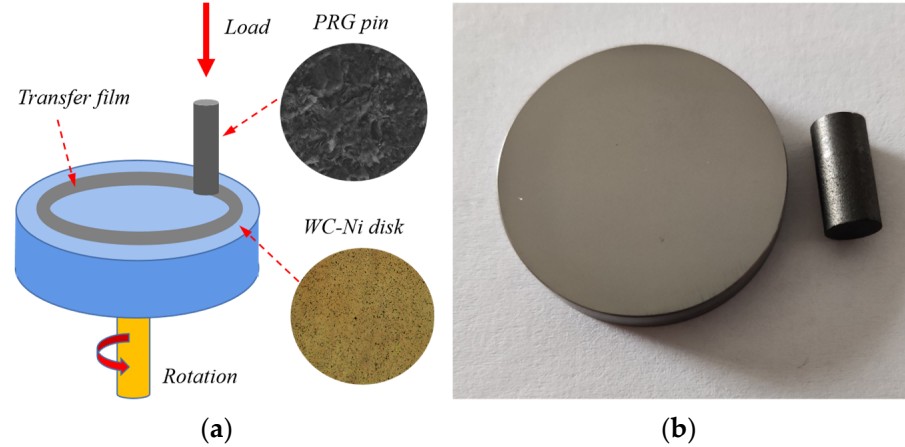

**Figure 1.** (**a**) Schematic diagram of sliding friction test; (**b**) Photo of PRG pin and WC-Ni disk.

## 3. Results and Discussion

### 3.1. Characteristics of PRG

Figure 2a shows the XRD and SEM images of fresh PRG. The characterization diffraction peaks at 26.54° and 54.6° were detected, and assigned to (002) and (101) crystallographic planes of graphite, respectively. The results show that graphite has the hexagonal phase structure and presents a multilayer microstructure, which is consistent with the SEM observation. The characterization diffraction peaks at 42.6°, 44.35° and 77.57° are assigned to (100), (101) and (110) crystallographic planes of graphite. It is concluded from the measurement of XRD that graphite has a certain orientation structure after high temperature hot pressing. According to energy-dispersive X-ray spectroscopy (EDS, Hitachi, Tokyo, Japan) analysis, it is believed that there are 96 wt.% of the carbon and 4 wt.% of the oxygen in graphite. A thermogravimetric analyzer (TA, NETZSCH, Shanghai, China) was utilized to estimate the thermal resistivity of the PRG as shown in Figure 2b, the PRG powder was heated in ambient air from 40 to 1000 °C at a ramp rate of 10 °C/min. Differential scanning calorimetry (DSC, NETZSCH, Shanghai, China) was employed to investigate the curing behavior of PRG. It can be seen from the curve that the weight of PRG powder at 200, 300 and 400 °C are 99.3%, 98.9% and 98.4%, respectively. The thermal weight loss below 150 °C is generally considered to be the removal of free water in PRG. The weight loss in the range of 150 to 300 °C is considered to be the removal of unreacted functional groups such as hydroxymethyl in phenolic resin. Finally, the weight loss in the range of 300 to 450 °C is mainly considered to be the thermal degradation of oxygen-containing functional groups such as phenolic hydroxyl, methylene and diphenyl ether. The heat resistance of PRG can be improved by etherification, esterification and strict curing conditions of phenolic hydroxyl groups.

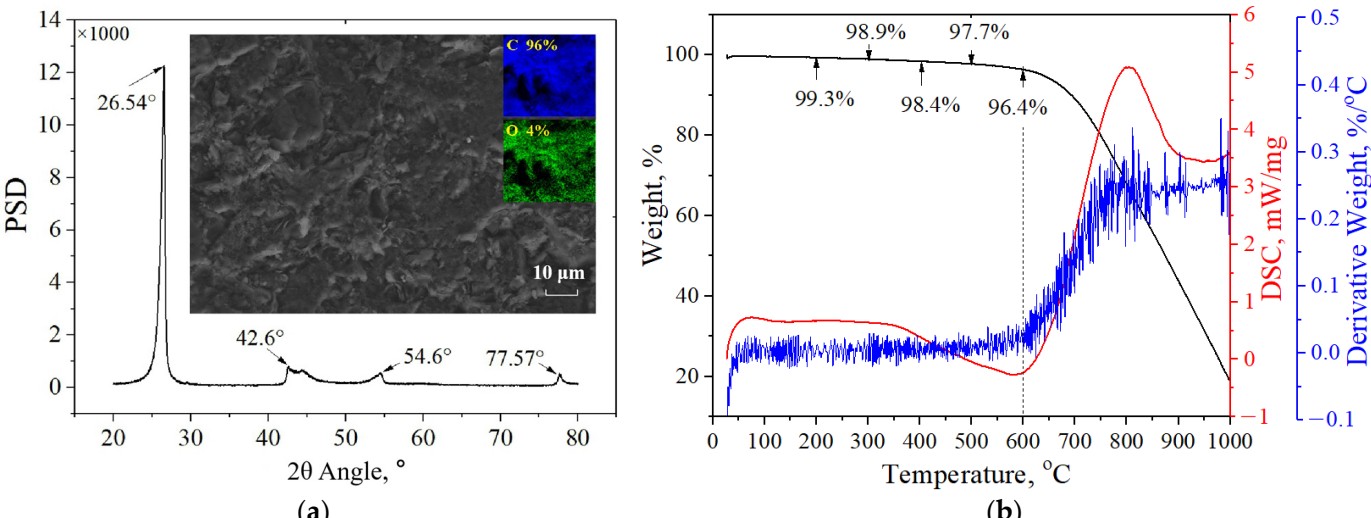

**Figure 2.** Characteristics of PRG: (**a**) XRD and SEM; (**b**) Weight loss and derivative weight loss curves.

### 3.2. High Temperature Tribological Properties of PRG

Figure 3 shows CoF as a function of sliding time for the friction pairs of PRG pin and WC-Ni disc under various temperatures of RT, 200, 300 and 400 °C. It can be seen that at room temperature, CoF increases slightly with the increase of test time, but the value of CoF is relatively stable. Although CoF of 0.15 at the initial stage increases to the maximum value of 0.19, the average CoF in the whole experimental period is about 0.17. However, it is found that the CoF curves at different temperatures are totally different, the running-in time and the duration of the lowest CoF are greatly affected by temperature. At 200, 300 and 400 °C, the unstable running-in time is 1500, 800 and 600 s, respectively ($T_{i200} > T_{i300} > T_{i400}$), and the duration of the low CoF is 2100, 1900 and 300 s, respectively ($T_{f200} > T_{f300} > T_{f400}$). It is shown that at 200 °C, the initial CoF of 0.114 increases to the maximum value 0.164 and then decreases to a stable value of 0.014 with a running-in time

of 1500 s. The average lowest CoF 0.014 remains stable in the rest of the experimental time of 2100 s. The CoF curve at 300 °C is different from that at 200 °C. The CoF increases from 0.06 to the maximum value 0.107 and then decreases to the lowest value 0.014 with a running-in time of 800 s, and then the CoF fluctuated and ranged from 0.02 to 0.04 in the following 1900 s. After that, CoF increases rapidly. Similarly, CoF curve at 400 °C is almost the same. The CoF increases from 0.114 to the maximum value 0.137 and then decreases to the value 0.012 with a running-in time of 600 s, and then the CoF remains a relatively stable value of 0.011 in a duration of 300 s. After that, CoF increases rapidly and remains at 0.07 finally.

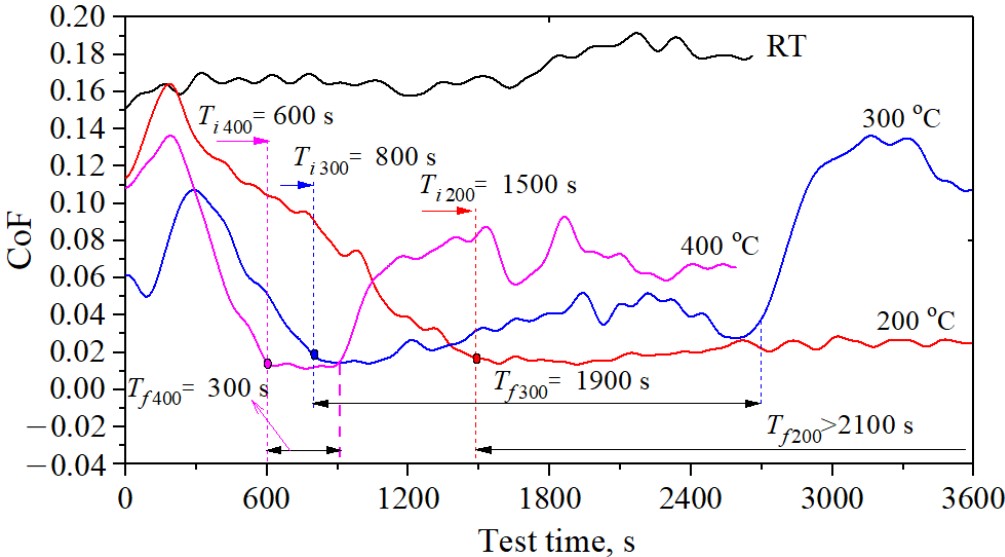

**Figure 3.** CoF of the friction pair of PRG/WC-Ni alloy under different temperatures.

Figure 4 shows SEM images and microscope images of the worn surface of WC-Ni disc under different temperatures. The W, Ni, C, and O elements in the red wire frame area of the friction surface were detected by the energy-dispersive X-ray spectroscopy (EDS). There is a high content of carbon on the worn surface. This confirms that the graphite was transferred to the surface of the cemented carbide disc from the PRG pin. At room temperature, as shown in Figure 4a,b, the transferred graphite films are distributed with uneven and irregular sizes and shapes, such as dots, flakes, and stripes, and the thickness of the transferred film is also uneven. At 200 °C, as shown in Figure 4c,d, the effective contact area between the pin and the disk is relatively small, but the transferred film is stable and continuous throughout the whole wear track. The graphite transferred film at 300 °C is shown in Figure 4e,f. The transferred film is mainly composed of small diameter granular graphite clusters, which are evenly distributed on the wear surface but the density of graphite clusters is low. As shown in Figure 4g,h, the graphite transferred film at 400 °C has a certain area and is continuous and sheet-like. Compared with the graphite transferred film at 300 °C, this sheet-like transferred film is formed by the adhesion of multiple layers of granular graphite clusters under high temperature and multiple interactions. Comparing the images at different temperatures, it can be seen that graphite was formed by a granular or flake-like transferred film on the surface of the cemented carbide under the combined action of high temperature and extrusion and shear stress. The difference is that graphite was formed as a continuous and compact transferred film with a certain thickness on the surface of the cemented carbide at 200 °C, which is conducive to achieving ultra-low friction. As the temperature raised, at 300 and 400 °C, graphite became granularly distributed on the surface of WC-Ni alloy, and a compact continuous transferred film was gradually formed on the surface of the alloy when friction time increased to a certain value.

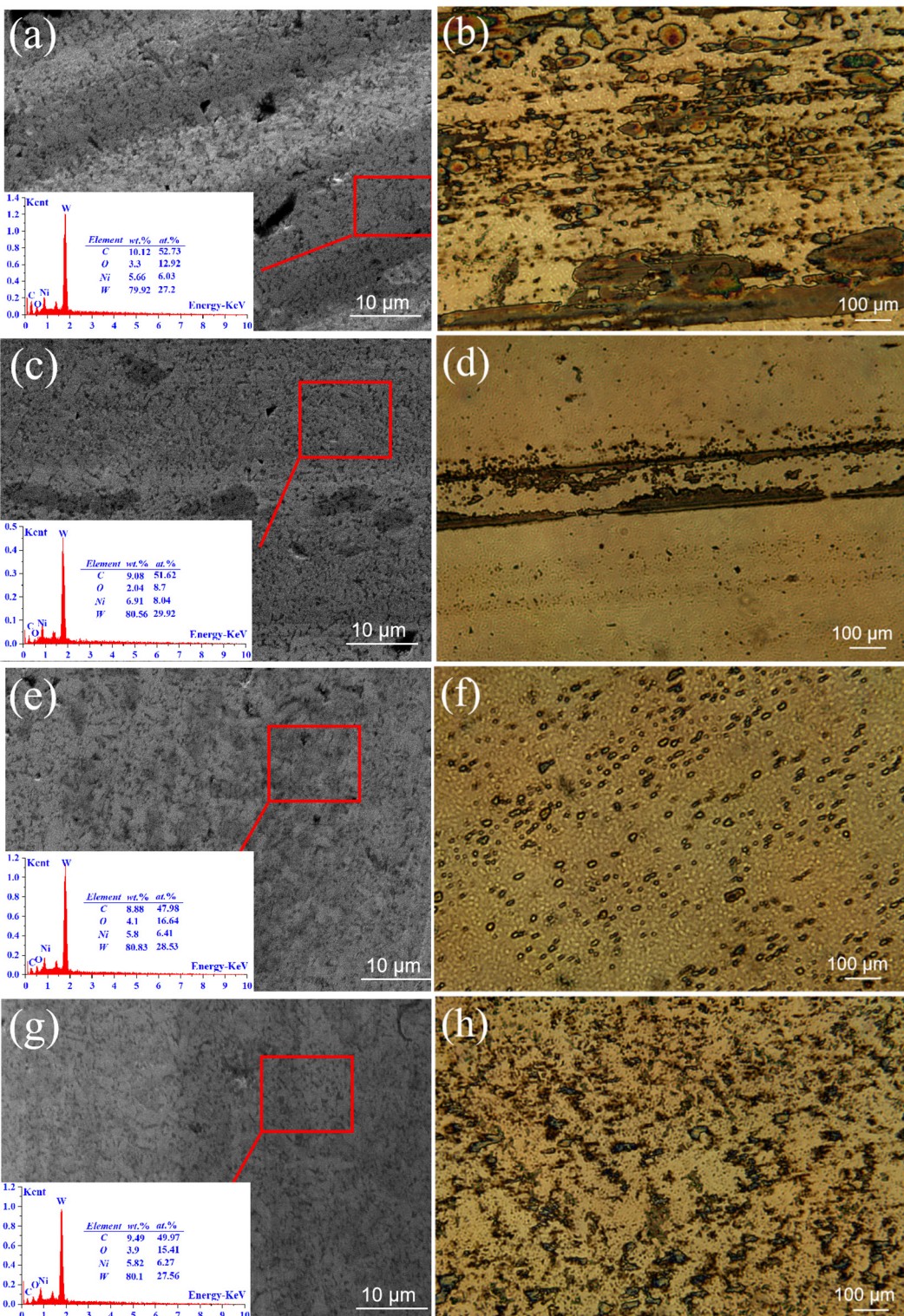

**Figure 4.** SEM and Microscope images of WC-Ni alloy discs. SEM images: (**a**) RT, (**c**) 200 °C, (**e**) 300 °C, (**g**) 400 °C; microscope:(**b**) RT, (**d**) 200 °C, (**f**) 300 °C, (**h**) 400 °C.

Figure 5 shows SEM images of the friction area of PRG pins after the tribotests at different temperatures. From the upper right corner to the lower left corner of each picture are the non-working area, the first contact area of friction, and the stable friction area. The yellow box area is the EDS element analysis area. It can be seen that there are different amounts of W and Ni in the worn surface, indicating that the WC-Ni alloy was worn and

transferred to the worn surface of the graphite pin during the friction process. The graphite material scraped the hard particles from the surface of the cemented carbide like a sharp blade. However, the wear surface of the WC-Ni alloy is covered with graphite transferred film and is difficult to be observed by SEM and microscope, but can be obtained from the element analysis of the friction surface of the PRG pin. At the same time, it can be seen from Figure 5 that the interaction process of the graphite pin and the WC-Ni alloy at different temperatures has the same and different points. At room temperature (Figure 5a), there are cracks in the first contact area of friction (such as area I). This is caused by extrusion and shear stress that acted on the graphite surface, and the graphite is crushed into irregular fragments and formed a transferred film on the surface of the cemented carbide (Figure 4b). This phenomenon exists in the first contact area of friction at 200, 300 and 400 °C. At the same time, it can be seen that there are white-gray spherical particles of different sizes in the friction area, and the mass fraction of W element in the spherical particles is found to account for 70 wt.% by the point scanning function of EDS. There is a certain difference of the friction process and state at 200 °C and room temperature. As shown in Figure 5b, there are not only large-sized spherical particles (about 4 μm in diameter) in the friction front area of graphite, but also smaller particles or nearly powdery alloy materials that existed in area II. There are more W and Ni elements in area II that can be seen from the EDS element distribution diagrams of area II and area III, with the mass fraction of W element in the spherical particles accounting for 59 wt.%. At 300 °C (Figure 5c), the powdery area, like area II in the friction front area of the graphite material, almost occupies the entire observation window. Compared with 200 °C, the area of the powdery area is larger and the size of the spherical particles is smaller (the diameter is less than 1 μm). There are some differences at 400 °C (Figure 5d), such as the W element in the friction area is no longer in the form of particles but in the form of unevenly distributed powder flakes, and the graphite is not particularly crushed in the friction front area. Compared with Figure 4f,h, it can be guessed that when the test temperature is above 300 °C, the graphite material will form micro-cracks on the surface due to extrusion and shear stress. Because the covalent bonds between graphite molecules are easily broken by shearing at high temperatures, graphite with cracked fracture is quickly cut into flakes and forms particles or even powder and adheres to the surface of cemented carbide.

### 3.3. Raman Spectroscopy

Figure 6 shows the Raman spectra of the PRG pin before and after the friction test under different temperatures. It is generally believed that the G peak reflects the symmetry and order of the material. Peak D characterizes the disorder of graphite, and the ratio of $I_D/I_G$ characterizes the defect density, which increases with the increase of the degree of disorder in the graphite structure. The 2D peak corresponds to the packing feature of graphite and reflects the ordered structure of graphite. The spectrum of the fresh surfaces has three typical peaks: D peak at 1344 cm$^{-1}$, G peak at 1580 cm$^{-1}$ and 2D peak at 2695 cm$^{-1}$. After the friction tests as shown in Figure 5, the D peaks are moving to a higher beam and approaching the standard grapheme beam of 1350 cm$^{-1}$. The 2D peaks at RT, 200, 300 and 400 °C are basically at 2700 cm$^{-1}$, and the intensities are 444, 554, 532 and 798 counts, respectively. The stacked structures of graphite at RT, 200 °C and 300 °C were weaker than that of the fresh graphite, and the stacked structure at 400 °C was stronger than that of the fresh graphite. Another feature of the Raman spectra was the shift of the G band. In this paper, the shift of the G band was caused by the friction between PRG and cemented carbide. The G peaks of the test graphite materials have shifted to a higher wavenumber by no more than 4 cm$^{-1}$ than the position of G peak of the fresh graphite, and the ratios of $I_D/I_G$ after abrasion tests at different temperatures are basically the same. Wang et al. [30] reported that the lower the crystallinity of the graphite sample, the higher the position of the G band. Compared with the G band positions of graphite after tests at 300 and 400 °C, the G band intensity of graphite at 200 °C is the lowest and the crystallization is the best.

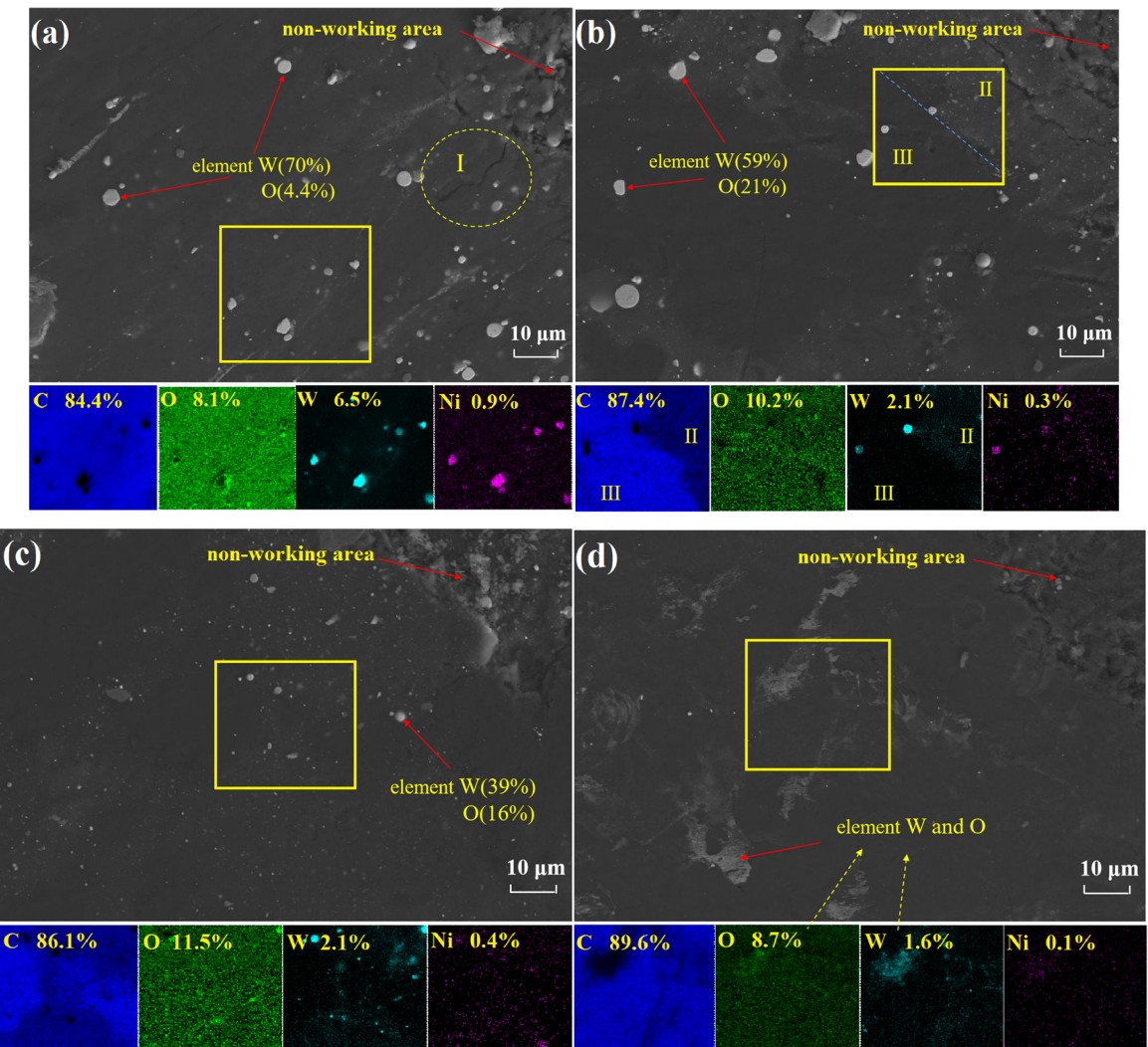

**Figure 5.** SEM of PRG pin under different temperatures: (**a**) RT; (**b**) 200 °C; (**c**) 300 °C; (**d**) 400 °C.

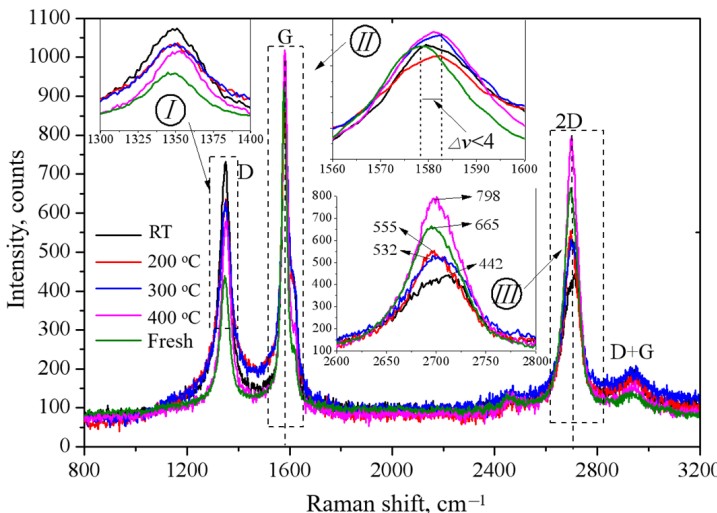

**Figure 6.** Raman spectra of the worn surface of PRG pin under different temperature.

Figure 7a shows Raman spectroscopy of the graphite transferred film on the worn surface of WC-Ni alloy. The D bands are ranged between 1343 and 1346 cm$^{-1}$, and the G bands are observed at 1590, 1601, 1600, and 1605 cm$^{-1}$, respectively, from RT to 400 °C.

Compared with the fresh graphite surface, the positions of the G peaks have shifted 10, 21, 20 and 25 cm$^{-1}$ to a higher beam direction, respectively, indicating that the order and symmetry of graphite have been significantly improved, especially with the improvement under high temperatures becoming more obvious. The 2D peaks are observed at 2680, 2687, 2689 and 2688 cm$^{-1}$ from RT to 400 °C, respectively, indicating that the stacking characteristics of the graphite transferred films at high temperature are the same, but there is a decrease compared to the original graphite (The 2D peak at 2695 cm$^{-1}$). The Raman spectrum of the graphite transferred film on the cemented carbide surface at 400 °C has some difference. As shown in Figure 7b, the Raman curve of the graphite tribofilms has a peak at 890 cm$^{-1}$, and the corresponding compound is C–O–C cyclopentane. This indicates that there was an oxidation reaction at this temperature and the graphite was partially oxidized.

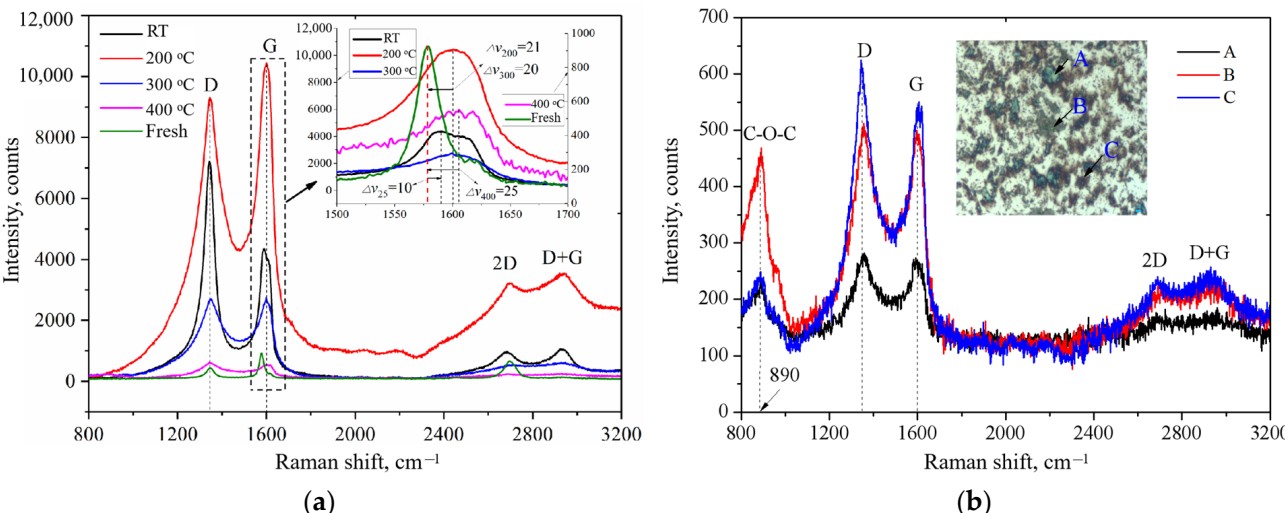

**Figure 7.** Raman spectra of the worn surface of WC-Ni alloy: (**a**) RT, 200 °C, 300 °C, 400 °C; (**b**) 400 °C.

### 3.4. High Temperature Friction Mechanism of PRG

In view of the above tribotest results and analysis, Figure 8 shows the transferred film formation mechanism of low coefficient of friction of PRG sliding against WC-Ni under dry friction at high temperature. As shown in Figure 3, The CoF curves at the beginning of the running-in stage show the characteristics of first increasing and then decreasing at 200–400 °C. This is because during the heating process, the surface of PRG material was softened due to heat, because there was no relative movement between the disc and the pin, no stable friction layer was formed on the surface of the friction pair. Therefore, at the running-in stage, under the action of squeezing force and shearing force, the rough peaks on the surfaces of the two counter-parts sheared each other and formed granular or flake-like fragments (Figure 8). This process is a running-in and unstable process, so the friction coefficient increases. When the surface of the PRG pin is compacted to form a continuous contact area with a large area and stable mechanical properties, and a stable and continuous surface is formed on the WC-Ni disc for the gap between the rough peaks of the material is filled with adhered PRG particles, the graphite gradually forms a stable transferred film on the surface of the metal disk, and the friction coefficient begins to decrease until it reaches a stable state. Under the action of high temperature and extrusion shear stress, the disordered graphite becomes an ordered layered structure and combines with the metal material with strong bond interaction. At room temperature, graphite materials are mainly subjected to a squeezing force and shearing force, and graphite is crushed to produce cracks and form uneven and irregular size and shape distributions such as dots, flakes, and stripes. The thickness of the transferred film is also different. At the same time, the local area in the point contact state generates high temperature and causes thermal adhesion, so the friction coefficient of the pair is relatively large and has a certain fluctuation.

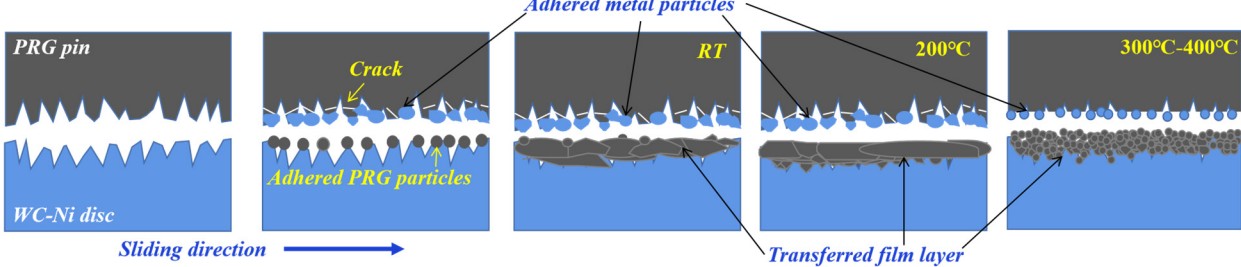

**Figure 8.** Schematic diagram of transferred film formation of PRG at high temperature.

The reason for the formation of stable low friction coefficient at 200 °C can be explained by the tribological properties of the friction materials and the effects of the morphology and chemical properties of the friction layer formed at the interface. It can be seen from Figure 5b that there are some cracks in the first friction contact area of the PRG pin, with no cracks observed elsewhere in the friction; therefore, it is considered that the cracks are caused by the force in the tribotests. The front of the first friction contact area is the non-working area, which will be contacted when the tribotest is further carried out. This non-working area may be uneven or grooved resulting from the adopted treatment, and it is in a stress-free state due to no work. Therefore, the stress difference will be generated under the action of directional extrusion and shear, and the graphite will move towards the non-working area. This is a reason for cracks during stable operation of the tribotests. In the initial stage of tribotest, the interaction of extrusion force and shear force between rough peaks of metal disk surface and rough peaks of PRG pin surface during the tribotest is another reason for the cracks. The graphite is stripped from the PRG pin by rough peaks of cemented carbide and a compact and continuous transferred film is formed on the surface of WC-Ni disk. Spherical particles containing W element are observed on the surface of PRG pin, and this should be because the hard particles with unstable bonding strength on the surface of cemented carbide are peeled off from the alloy surface under the action of shear force and embedded into the graphite surface under the action of extrusion force. In the process of force and mutual movement, the alloy particles in graphite peel off the cemented carbide surface, forming a continuous process of mutual peeling and embedding. This also leads to the long-term stability of the coefficient of friction after the running-in stage. The compaction of friction contact interface and the growth of friction path in the particles containing W element will be extruded into smaller particles or close to powder. In addition to graphite, there are phenolic resin, coke and organic fiber in the PRG pin. At 200 °C, the tribological properties of PRG are affected by the chemical properties of oxygen and $H_2O$ molecules inherent in graphite matrix and the external test environment. During the tribological test, the active sites of the carbon network at the graphite contact peak have high energy and can carry out multiple high-speed reactions. The saturation of the ambient atmosphere to the dangling bond leads to the formation of the graphite lubrication layer. These high-energy activation energies may come from the mechanical energy provided in the tribological process [28]. In this case, dissociative chemisorption of $H_2O$ on carbon atoms located in a vacancy leads to the formation of hydroxyl [31]. At the same time, a large number of dangling bonds will interact with oxygen in the atmosphere and water in graphite, mainly producing oxygen-containing groups and hydrogen carbon bonds (C–OH and C–H).

In the tribotests at 300–400 °C, the friction coefficient began to rise after a period of stability in the ultra-low zone. This should be due to the thermal degradation and partial oxidation of phenolic resin in the PRG [32]; the oxide and thermal adhesion between the friction pair materials lead to increased friction coefficient. When PRG is heated above 300 °C in air, the water molecules are gradually eliminated, which is attributed to the condensation reaction between hydroxymethyl in phenolic resin and hydrogen in other organic compounds. The polymers is oxidized by oxygen in the atmosphere to form diphenyl methanol substances and bonds. These substances would lead to the

formation of oxidised terminal groups, such as aldehydes and carboxylic acids, and also undergo oxidation branching. A vast majority of unstable carboxylic groups and hydroxylic groups are eliminated by thermal decarboxylation and transformed into more thermally stable lactones, anhydrides, ethers and carbonyls. Most of C–O and O–H components are removed by intramolecular dehydration, and the C–O–C compounds found in Figure 7b are attributed to a phenolic O–H···O–H condensation reaction. The oxide generated in the thermosetting process of polymers such as phenolic resin fills the graphite crystal space and improves the strength and compactness of graphite tribofilms. Under this thermal degradation, the thermally softened material on the PRG surface is cut off by the relative moving shear force and attached to the cemented carbide surface in the form of small particles. Therefore, it is difficult to observe obvious crack structure in the friction front area of graphite contact surface. Before the formation of compact transferred film, metal surface materials will exist on the surface of graphite materials in the form of smaller particles or even powder for the comprehensive action of thermal adhesion and shear force (Figure 5). The interaction between the graphite surface containing WC particles and the graphite tribofilms on the WC-Ni disk will form stable tribological behavior. The thermal softening of the material and the generated oxide at high temperature will increase the thermal adhesion between graphite layer and cemented carbide, increasing the friction coefficient. This may be the reason why the CoF increases rapidly from 0.011, which was a relatively stable value in a duration of 300 s at 400 °C. It can be predicted that when the friction time increases to a certain value, the granular graphite transferred film will gradually overlap, the compact and continuous transferred film will be gradually formed on the surface of cemented carbide, and the coefficient of friction will gradually become stable after oxide and graphite form a stable lubricating film.

### 4. Conclusions

The tribological properties of the friction pair of the PRG pin and WC-Ni alloy disc at different temperatures (RT, 200, 300 and 400 °C) under a normal load of 10 N and a speed of 100 mm/s were investigated systematically in ambient air. The conclusions are summarized as follows:

(1) The PRG material can achieve ultra-low friction coefficient of 0.01–0.015 at 200, 300 and 400 °C. The differences are that with the increase of test temperature, the unstable running-in time to achieve the low friction coefficient decreases and the maintenance time of the low friction coefficient decreases.

(2) At room temperature, the graphite material is mainly subjected to extrusion and shear force and crushed to form cracks, forming point, sheet and strip fragments with uneven size and shape. At 300 and 400 °C, the graphite tribofilms are formed and characterized by distributed granular graphite and layer by layer superposition. At 200 °C, the graphite material can form a compact and continuous transferred film and the friction coefficient is stable for a long time.

(3) Under the combined action of thermal adhesion and shear force, graphite is formed at the transferred films on the surface of cemented carbide, and metal surface materials will also exist on the surface of graphite materials. The interaction between the graphite surface containing WC particles and the graphite tribofilms on the WC-Ni disk will form stable tribological behaviors. The oxide generated in the thermosetting process of polymers can fill the graphite crystal space and improve the strength and compactness of graphite tribofilms.

**Author Contributions:** Conceptualization, F.Z. and Q.Z.; data analysis, F.Z., P.Y. and Q.Z.; investigation, F.Z., P.Y. and Q.Z.; writing-original draft, F.Z., P.Y. and Q.Z.; writing-review and editing, F.Z., P.Y. and Q.Z.; project administration, F.Z. and J.W.; funding acquisition, F.Z. and Q.Z. All authors have read and agreed to the published version of the manuscript.

**Funding:** This work was financially supported by the National Natural Science Foundation of China (Grant No. 51805409 and 51675409), Doctoral Research Fund Project of Taiyuan University of Science

and Technology (Grant Nos. 20202010 and 20212013), Scientific and Technological Innovation Programs of Higher Education Institutions in Shanxi (Grant No.2020L0357), National Science and Technology Major Project (j2019-IV-0004-0071) and the Natural Science Foundation of Chongqing, China (cstc2019jcyj-msxmX0577).

**Institutional Review Board Statement:** Not applicable.

**Informed Consent Statement:** Not applicable.

**Data Availability Statement:** Data is contained within the article.

**Conflicts of Interest:** The authors declare no conflict of interest.

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
