# Peer review of "Insights on the Formation Mechanism of Ultra-Low Friction of Phenolic Resin Graphite at High Temperature"

_coatings, doi:10.3390/coatings12010006_

Round 1
Reviewer 1 Report
Despite the efforts done by the authors, and the manuscript contains a good scientific story, there are some important points that should be considered by the authors.
- Abstract
- Line 15: contact interface at different temperatures (200oC, 300oC and 400oC). Change to contact interface at different temperatures (room temperature, 200 oC, 300 oC and 400 oC). RT condition should be added as mentioned in experimental and results
- Clarify what is the abbreviation, especially when the symbol is mentioned the first time: Line 11 WC-Ni
- Introduction
- Clarify what is the abbreviation, especially when the symbol is mentioned the first time: Line 29 MoS2 should be changed to molybdenum disulphide (MoS2)
- line 29: Compared with traditional basic lubricants such as MoS2, please change to ……………… such as MoS2.
- Add the following two references to Reference No. [17] Line 61…. [17-19]
- [18] Microstructure and mechanical properties of hot extruded 6016 aluminum alloy/graphite composites,
https://doi.org/10.1016/j.jmst.2018.03.004
- [19] Strength and Wear Behavior of Mg Alloy AE42 Reinforced with Carbon Short Fibers, https://doi.org/10.1007/s40195-018-0771-z
- Please add space between the numbers and its units, such as Line 73; 700oC, 800oC and 900oC ….. should be changed to 700 oC, 800 oC and 900 oC and please check the whole manuscript.
- One format should be considered in writing all authors’ names. Sometimes you write given name and surname such as in line 75: (Kumar N et al. [24]), line 80-81: (Lafon - Placette S et al. [25]) and several times you write only the surname (the correct form that should be considered) such as inline 83; Zhao et al. [17]
- Experimental
- Line 102 and line 103 Please add the compacting pressure and the sintering conditions (Temperature and time) for the produced WC/Ni alloy disk.
- Results
- Use one type of parentheses when writing axes for all Figures (for example, the authors used (---) in figure 2 and [---] in figures 3, 6, and 7)
- Scale bar should be added for OM images in figure 4 instead of x magnification.
- Conclusions
- The conclusions should be rewritten in specific points that contain the most important results without explanation

Author Response
Dear Editor and Referee,
Thank you very much for the useful comments and valuable suggestions on our manuscript. We have studied the comments from the reviewers carefully and made corrections which we hope to meet with approval. Revised portions are marked in red in the manuscript. The responses to the reviewer’s comments are as following:
Comments and Suggestions for Authors (Reviewer 1)
Despite the efforts done by the authors, and the manuscript contains a good scientific story, there are some important points that should be considered by the authors.
For “Abstract”:
1. Line 15: contact interface at different temperatures (200oC, 300oC and 400oC). Change to contact interface at different temperatures (room temperature, 200 oC, 300 oC and 400 oC). RT condition should be added as mentioned in experimental and results.
Response: Thanks for your concerning. “room temperature” has been added and the sentence has been changed to “...contact interface at different temperatures (room temperature, 200 oC, 300 oC and 400 oC)”.
2. Clarify what is the abbreviation, especially when the symbol is mentioned the first time: Line 11 WC-Ni.
Response: To solve the problem of abbreviation for the first time, in the abstract, the original description has been changed from "...WC-Ni alloy" to "...tungsten carbide-nickel (WC-Ni) alloy"
For “Introduction”:
3. Clarify what is the abbreviation, especially when the symbol is mentioned the first time: Line 29 MoS2 should be changed to molybdenum disulphide (MoS2)
Response: To solve the problem of abbreviation for the first time, the original description has been changed from "...MoS2" to "...molybdenum disulphide (MoS2)"
4. line 29: Compared with traditional basic lubricants such as MoS2, please change to ……………… such as MoS2.
Response: Thanks for your concerning. "...MoS2" has been changed to "...MoS2"
5. Add the following two references to Reference No. [17] Line 61…. [17-19]
[18] Microstructure and mechanical properties of hot extruded 6016 aluminum alloy/graphite composites,https://doi.org/10.1016/j.jmst.2018.03.004
[19] Strength and Wear Behavior of Mg Alloy AE42 Reinforced with Carbon Short Fibers, https://doi.org/10.1007/s40195-018-0771-z
Response: This paper has cited these two references as [18] and [19] and another two references as [22] and [23] in the main body and references, and made corresponding changes to other information in the manuscript.
6. Please add space between the numbers and its units, such as Line 73; 700oC, 800oC and 900oC ….. should be changed to 700 oC, 800 oC and 900 oC and please check the whole manuscript.
Response: The whole manuscript has been checked and space has been added between all the numbers and its units.
7. One format should be considered in writing all authors’ names. Sometimes you write given name and surname such as in line 75: (Kumar N et al. [24]), line 80-81: (Lafon - Placette S et al. [25]) and several times you write only the surname (the correct form that should be considered) such as inline 83; Zhao et al. [17]
Response: All authors’ names in the references in the introduction have been uniformly formatted, and only surnames are used.
For “Experimental”:
8. Line 102 and line 103 Please add the compacting pressure and the sintering conditions (Temperature and time) for the produced WC/Ni alloy disk.
Response: The compacting pressure and the sintering temperature for the produced WC-Ni alloy disk have been added in line 106 and line 107. The compaction pressure was greater than 2200 MPa and the maximum sintering temperature was 1450 oC.
For “Results”:
9. Use one type of parentheses when writing axes for all Figures (for example, the authors used (---) in figure 2 and [---] in figures 3, 6, and 7)
Response: (---) is used as a unified type of parentheses when writing axes for all figures such as Figure 2, Figure 3, Figure 6 and Figure 7.
10. Scale bar should be added for OM images in figure 4 instead of x magnification.
Response: Scale bar has been added for microscope images in Figure 4b, Figure 4d, Figure 4f and Figure 4h instead of X magnification.
For “Conclusions”:
11. The conclusions should be rewritten in specific points that contain the most important results without explanation
Response: Thanks for your concerning. The conclusions have been rewritten in specific points that contain the most important results without explanation. The revised conclusions have been marked in red.
We have tried our best to improve the manuscript and made some changes in it. These changes will not influence its content and framework. We look forward to your further response. Thanks again for your helpful and encouraging comments!
With best regards.
Yours sincerely,
Fan Zhang, Peng Yin, Qunfeng Zeng, Jianmei Wang

Reviewer 2 Report
Presented work is very interesting and valuable. The work is written carefully and does not require any substantive corrections. However, please explain at work how "wt%" was calculated. Moreover, on what basis was it assumed that there are cracks on the surfaces and not grooves or unevenness resulting from the adopted treatment (diagram Fig. 8)?
Author Response
Dear Editor and Referee,
Thank you very much for the useful comments and valuable suggestions on our manuscript. We have studied the comments from the reviewers carefully and made corrections which we hope to meet with approval. Revised portions are marked in red in the manuscript. The responses to the reviewer’s comments are as following:
Comments and Suggestions for Authors (Reviewer 2)
Presented work is very interesting and valuable. The work is written carefully and does not require any substantive corrections. However, please explain at work how "wt%" was calculated. Moreover, on what basis was it assumed that there are cracks on the surfaces and not grooves or unevenness resulting from the adopted treatment (diagram Fig. 8)?
Response for “please explain at work how "wt%" was calculated”: The energy-dispersive X-ray spectroscopy (EDS) module equipped with scanning electron microscopy (SEM, su-8010) has its own software, which can automatically identify the composition and mass percentage of elements during point scanning. Although there is error, it has a certain reference. The "wt%" at this work was obtained from EDS during point scanning.
Response for “on what basis was it assumed that there are cracks on the surfaces and not grooves or unevenness resulting from the adopted treatment (Diagram Fig.8)”: The cracks mentioned in this paper are observed in the first friction contact area of PRG pin after tribotests, which can be seen from Figure 5. And no cracks are observed elsewhere in the friction area. Therefore, it is considered that the cracks are caused by the force in the tribotests. The front of the first friction contact area is the non-working area, which will be contacted when the tribotest is further carried out. This non-working area may be uneven or grooved resulting from the adopted treatment, and it is in a stress free state due to no work. Therefore, the stress difference will be generated under the action of directional extrusion and shear, and graphite will move towards the non working area. This is a reason for cracks during stable operation of the tribotests. In the initial stage of friction test, the interaction of extrusion force and shear force between rough peaks of metal disk surface and rough peaks of PRG pin surface during the tribotest is another reason for the cracks. This explanation has been added in Section 3.4 of the manuscript and marked in red.
We have tried our best to improve the manuscript and made some changes in it. These changes will not influence its content and framework. We look forward to your further response. Thanks again for your helpful and encouraging comments!
With best regards.
Yours sincerely,
Fan Zhang, Peng Yin, Qunfeng Zeng, Jianmei Wang
